# A Comprehensive Study on the Electrostatic Properties of Tubulin-Tubulin Complexes in Microtubules

**DOI:** 10.3390/cells12020238

**Published:** 2023-01-05

**Authors:** Wenhan Guo, Tolulope Ayodeji Ale, Shengjie Sun, Jason E. Sanchez, Lin Li

**Affiliations:** 1Computational Science Program, University of Texas at El Paso, El Paso, TX 79902, USA; 2Department of Physics, University of Texas at El Paso, El Paso, TX 79902, USA

**Keywords:** microtubule, tubulin, protein-protein interactions, molecular dynamics simulation, DelPhi, DelPhiForce, electrostatic features, salt bridges, hydrogen bonds

## Abstract

Microtubules are key players in several stages of the cell cycle and are also involved in the transportation of cellular organelles. Microtubules are polymerized by α/β tubulin dimers with a highly dynamic feature, especially at the plus ends of the microtubules. Therefore, understanding the interactions among tubulins is crucial for characterizing microtubule dynamics. Studying microtubule dynamics can help researchers make advances in the treatment of neurodegenerative diseases and cancer. In this study, we utilize a series of computational approaches to study the electrostatic interactions at the binding interfaces of tubulin monomers. Our study revealed that among all the four types of tubulin-tubulin binding modes, the electrostatic attractive interactions in the α/β tubulin binding are the strongest while the interactions of α/α tubulin binding in the longitudinal direction are the weakest. Our calculations explained that due to the electrostatic interactions, the tubulins always preferred to form α/β tubulin dimers. The interactions between two protofilaments are the weakest. Thus, the protofilaments are easily separated from each other. Furthermore, the important residues involved in the salt bridges at the binding interfaces of the tubulins are identified, which illustrates the details of the interactions in the microtubule. This study elucidates some mechanistic details of microtubule dynamics and also identifies important residues at the binding interfaces as potential drug targets for the inhibition of cancer cells.

## 1. Introduction

Microtubules are cytoskeletal structures that play an important role in shaping eukaryotic cell structure [1,2]. They are responsible for the movement of cellular components such as chromosomes, the mitotic spindle, and other organelles within the cell [3,4]. Microtubules are rigid, hollow rod structures (approximately 25 nm in diameter) that undergo growth and shrinkage continually within the cell [5,6]. Microtubules are composed of alpha (α) and beta (β) tubulin dimers [7,8]. Tubulin dimers form 13 protofilaments that bind laterally to construct cylindrical microtubules [7,9]. Tubulins are generally arranged in a head-to-tail formation, with β tubulin exposed at the plus end and α tubulin exposed at the minus end. This configuration determines the direction of movement of cytoskeleton molecular motors (kinesin and dynein) along the microtubule [10,11,12,13].

Currently, it is known that kinesin motors bind certain cargo and move along the microtubules in anterograde axonal transport [14,15]. Yet there is little understanding of how some materials are targeted to the axon. Abnormalities in the transport mechanism along the microtubules have been linked to some neurodegenerative diseases and cancer growth [16,17,18]. A defect in the transport mechanism of microtubules may complicate a variety of human diseases or play an important role in preventing recovery [19,20]. Hence, it is critical to understand the dynamics, including the composition, assembly, and disassembly, of microtubules [21,22]. The basic structural unit of microtubules (α and β tubulin dimers) contains two unidentical binding sites: the N-site (non-exchangeable) on α–tubulin binds GTP (guanosine triphosphate) molecules and the E-site (exchangeable) at β-tubulin can bind either GDP (guanosine diphosphate) or GTP [23]. Protein-protein docking and Molecular Dynamics (MD) simulations can help elucidate the features of α/β tubulins which are responsible for microtubule dynamic behavior. MD simulations of tubulins have revealed that interface flexibility is significant to tubulin assembly [24].

Computational methods have been widely used to study the functions of biomolecules [25,26,27,28,29] including tubulins. Dr. Alberto Redaelli’s group evaluated the elastic constants of an α-/β-tubulin monomer and the interaction force between the components by means of molecular dynamics simulations, and constructed a mechanical model of the tubulin dimer [30]; Joshi and Dima studied the microscopic origins of the mechanical response in microtubules by probing features of the energy landscape of the tubulin monomers and tubulin heterodimer with molecular simulations [31]; Marracino’s group explored the response of tubulin to strong electric fields at the nanosecond level in molecular dynamics simulations [32]; Dr. Aristide Dogariu’ lab reported accurate optical measurements of tubulin polarizability in aqueous suspensions [33]; Andriy Kovalenko et al. studied the microtubule stability by using the three-dimensional molecular theory of solvation [34]. Notably, there are many studies that focus on the electrostatic properties of tubulins. Dr. Satarić et al. presented the results of molecular dynamics computations based on the atomic structure of tubulin and explored the electrostatic properties of microtubules in 2005 [35]. The results included the values of net charge, charge distribution, and dipole moment components for the tubulin heterodimer. They summarized that the charge and dipole forces play significant roles in the formation of microtubules and the interactions among microtubules and other proteins. Malinski’ lab studied the tubulin alpha-beta heterodimer in terms of the test charges, test dipoles, and neighboring microtubules. They demonstrated the importance of electrostatics in the formation of the microtubule and the tubulin-kinesin binding strength [36]. Dr. Lin Li’s research work in 2016 investigated the interactions between microtubules and dynein. This work revealed that microtubules generate electrostatic funnels that guide the dynein’s microtubule-binding domain (MTBD) to dock at a precise, critical binding site on the microtubule. This work also demonstrated that long-range electrostatic interactions bring a degree of precision to help dynein’s stepping process [36]. Tuszynski et al. discussed microtubules and their electrostatic properties in a review paper, which focused on revealing how these biopolymers involve in a broad spectrum of intracellular electrical signaling [37]. In this work, we utilized multi-scale approaches to focus on the electrostatic features of tubulin-tubulin interactions.

Hydrogen bonds are significant to the stability of microtubules. Ayoub et al. (2014) revealed that lateral hydrogen bonds are significantly stronger than longitudinal ones [38]. However, this could be due to the presence of a GTP cap in β tubulin. Notwithstanding, electrostatic interactions are expected to stabilize the interactions in the longitudinal orientation more strongly than that of the lateral one. Our study is focused on tubulin-tubulin electrostatic interactions at four binding interfaces in the absence of a binding agent. In our work, we studied the electrostatic features, including the electrostatic potential, electric field lines, and electrostatic forces to better characterize the interaction among tubulins. Hydrogen bonds and salt bridges were also analyzed at different interfaces in the tubulin-tubulin complexes based on MD simulations.

## 2. Methods

### 2.1. Structure Preparation

In this work, we took two tubulin-containing structures (PDB ID: 6TA4 [39] and PDB ID: 5MLV [40]) as templates and modeled a 4 × 2 array of tubulins as shown in Figure 1. The tubulin proteins were in the adenylyl imidodiphosphate (AMP-PNP) state. In the entire microtubule structure, at total of four binding interactions were identified. These include: α tubulin binding to α tubulin (α/α tubulin complex), β tubulin to β tubulin (β/β tubulin complex), α tubulin to β tubulin (α/β tubulin complex), and β tubulin to α tubulin (β/α tubulin complex).

### 2.2. Electrostatic Potential

We calculated the electrostatic potential of the tubulins using DelPhi [41,42]. DelPhi calculates the electrostatic potential by using a finite difference method to solve the Poisson-Boltzmann equation (PBE) for biomolecules.
(1)$$\nabla \cdot \left[\epsilon \left(r\right)\nabla \phi \left(r\right)\right]=-4\pi \rho \left(r\right)+\epsilon \left(r\right)\kappa^{2}\left(r\right)\sin h\left(\phi \left(r\right)/k_{B}T\right)$$
where ϕ(r) is the electrostatic potential, ρ(r) is the permanent charge density based on the atomic structures, ε(r) is the dielectric permittivity, κ is the Debye-Hückel parameter, *k_B_* is the Boltzmann constant, and *T* is temperature.

For the Delphi calculation, the dielectric constants were set at 2.0 for protein and 80.0 for the water environment. The protein filling percentage of the Delphi calculation box was set to 70.0, the probe radius for the molecular structure was 1.4 Å and the salt concentration was set to 150 mM. The boundary condition for the Poisson Boltzmann equation was set to the dipolar boundary condition. DelPhi provides a number of options for modeling different types of ions [43]. Divalent cations are crucial for tubulin-tubulin interactions. To model the divalent cations, the multi-valence function was used and the valence of cations was set as +2. All the DelPhi parameters are discussed in described in the DelPhi manual (http://compbio.clemson.edu/lab/delphisw/) accessed on 31 December 2022. The calculated electrostatic surface potential was visualized with USCF Chimera [44] as shown in Figure 2. Positively and negatively charged regions are colored in blue and red, respectively. The color scale ranges from −1.0 to 1.0 kT/e.

### 2.3. Electric Field Lines

To see the strength of electric fields, Visual Molecular Dynamics (VMD) [45] was utilized. The tubulins were separated by 20 Å to improve the visualization of the field lines in our area of interest (Figure 3). The density of the electrostatic field lines indicates the strength of the electrostatic interactions between the tubulins.

### 2.4. Electrostatic Forces

To compare the strengths and directions of electrostatic forces among tubulins, DelphiForce [46,47] was used. The net forces calculated by DelphiForce were visualized with VMD and represented by blue arrows. Forces are shown at different distances from 14 to 40 Å with a step size of 2 Å. The tubulin/tubulin complexes were separated in the direction of the line that connects their centers of mass [48]. For better visualization of forces in VMD [45], arrows were normalized to the same size at variable distances so that only the direction of each force—and not its associated magnitude—is represented. The force direction trend is illustrated in Figure 4 and the magnitudes of the total electrostatic forces are visualized in Figure 5.

In this study, DelPhi package (including DelPhiForce) was utilized to calculate the electrostatic forces. Delphi package has been proved to be very successful at calculating the electrostatic features for protein-protein interactions. However, Delphi doesn’t perform simulations to consider flexibilities for the proteins. Therefore, for electric field lines and electrostatic potential calculations, tubulins were separated 20 angstroms away from each other to avoid clashes and computing errors when they are too closed. Moreover, at 20 angstroms distance, the Van der Waals forces decrease much more significantly than the electrostatic forces; thus, the electrostatic force is the dominant force in this situation.

### 2.5. Molecular Dynamic Simulations

Flexibility at the binding interfaces is crucial to study the interactions between tubulins. Therefore, four 10 ns simulations were performed on each of the tubulin complexes. In each simulation, the binding interfacial residues are free to move, but all other residues were not. This was done because salt bridges and hydrogen bonds are sensitive to the distance and angle of the pairwise residues.

The interactions at the binding interfaces of α/α-tubulin, β/β-tubulin, α/β-tubulin, and β/α-tubulin complexes were explored with NAMD [49] using an explicit solvent model and were carried out on Stampede2 at the Texas Advanced Computing Center (http://www.tacc.utexas.edu) accessed on 31 December 2022. The CHARMM force field [50] was used with the temperature set at 300 K and the pressure following Langevin dynamics. The minimization was set to 10,000 steps for each simulation. In each simulation, residues with any atom within 15 Å from the binding interfaces were treated as interfacial residues. All the interfacial residues were set free, while non-interfacial residues were fixed. The simulations were visualized by VMD (see Appendix A). Root Mean Square Deviation (RMSD) analysis shows the simulated structures are stable during the 10 ns MD simulations. We extracted one thousand frames from the last 5 ns for further analysis. To study the role of individual residues on binding interactions, we calculated the number of hydrogen bonds that exist within a 3.2 Å distance using the hydrogen bond tool in VMD. Additionally, we analyzed salt bridges that were formed within 4.0 Å in each binding interface by calculating their percent occupancy during the MD simulations. Salt bridges were identified by using the salt bridge tool in VMD. The interfacial residues that contribute significantly to the binding interactions were identified and explored further.

## 3. Results and Discussions

### 3.1. Surface Electrostatic Potentials of Tubulins

The structures and electrostatic potential surfaces of four pairs of tubulin complexes (α/α, β/β, α/β, β/α) are shown in Figure 2. DelPhi [was utilized to calculate the electrostatic potential on the surfaces of different tubulin complexes. The negatively and positively charged regions are red and blue, respectively. The color scale is set from −1 kT/e to 1 kT/e. From Figure 2A–D, we found only a few residues interacting with each other on the interfacial surfaces between α/α tubulins or between β/β tubulins. Yet the interfacial regions of α/β or β/α tubulins are very large (Figure 2E–H). The tubulin complexes are attracted to each other if their interfacial have opposite net charges; moreover, the attractive force is suspected to enhance the stabilities of the resulting complexes. The electrostatic potential of the tubulins indicates that the interfacial area between α/β tubulins has the largest surface area of interacting opposite charges of the four modes. The next largest is the interacting area between β/α tubulins and it is shown that there are some neutrally charged regions on β tubulin (Figure 2H, represent by white color), which causes the electrostatic interaction between β/α tubulin to be weakened. On the interfacial surfaces between α/α tubulins or between β/β tubulins, the regions with opposite charges at the corresponding positions are relatively small.

### 3.2. Electric Field Lines of Tubulins

The charge distribution and electric field lines in each binding mode show an interesting pattern. The α/α tubulin interface has weak repulsive interactions between tubulins in the complex. The electric field lines coming from both α tubulins repel each other (as shown in Figure 3A,E). This may explain why the dissociation occurs between parallel protofilaments. The binding interface of β/β tubulins is attractive only in a small area. It shows that very few electric field lines are involved in the interaction (as shown in Figure 3B,F), which is generated by a relatively small attractive electrostatic binding potential. At the α/β tubulin complex (as shown in Figure 3C,G), the binding interface of α tubulin is predominantly positively charged, while the binding interface of β tubulin is predominantly negatively charged. The α/β-tubulin interface produces the strongest electrostatic binding interaction among all the four possible binding interfaces in microtubules. This is evident by the very dense electric field lines involved in the binding interface (Figure 3C,G). Finally, the electric field lines at the β/α-tubulin interfaces show weak attractive interaction (as shown in Figure 3D,H). The strong binding interaction in the α/β-tubulin dimer and weak binding interaction in the β/α-tubulin dimer help explain why free tubulins always form α/β dimers.

### 3.3. Electrostatic Forces of Tubulins

Electrostatic forces of tubulin/tubulin complexes were calculated by DelPhiForce [46,47] (as shown in Figure 4 and Figure 5). Arrows in Figure 4 are illustrated to visualize the net forces acting on a selected tubulin when it is shifted away from the other tubulin at distances ranging from 14 Å to 40 Å with a step size of 2 Å. The arrow directions indicate the net force directions. Figure 4 shows that β/β, α/β, and β/α complexes have attractive forces at distances ranging from 14 Å to 40 Å because the overall direction trends are attractive. For the α/α tubulin complex, the overall trend of the blue arrows is repulsive. Thus, the interaction between α tubulin and α tubulin is repulsive, which is consistent with our electrostatic potential results. To illustrate the directions clearly, the sizes of the arrows in Figure 4 do not indicate the magnitudes of the forces. The magnitudes of the forces are shown in Figure 5.

To demonstrate the net force strengths between tubulins, we plotted the values of the electrostatic forces from the DelphiForce calculations shown in Figure 5. The trends in Figure 5 show that the electrostatic binding force decreases when the distance increases, which is consistent with Coulomb’s law. We also noticed that at 40 Å, the electrostatic binding force is very close to 0 kT/Å, which means that at this distance, the tubulins tend not to bind anymore. From Figure 5, the lowest net forces were found for the α/α tubulin complex. Combined with the results shown in Figure 4, we conclude that the interaction between α tubulin and α tubulin is weakly repulsive. In addition, the most attractive forces are those between α tubulin and β tubulin. Therefore, the free α tubulins and β tubulins tend to form α/β tubulin dimers.

### 3.4. Molecular Dynamics Simulations

A 10 ns simulation was performed for each of the tubulin/tubulin complexes. The RMSD plot for each simulation is shown in Figure 6. The last 5 ns of the simulations were selected for analysis because their structures are stable as shown in the RMSD plot (Figure 6). The simulations were visualized by VMD [45] (shown in Appendix A).

In our analysis, the salt bridges and hydrogen bonds which were rarely formed (<10% occupancy) were ignored. The residues involved in the hydrogen bonds and salt bridges > 10% occupancies were analyzed.

### 3.5. Hydrogen Bonds

The threshold distance for hydrogen bonds between the donor and acceptor was set to 3.2 Å while the angle cut-off was 20°. Hydrogen bonds at the binding interfaces of α/α, α/β, β/α, and β/β tubulin complexes were calculated based on MD simulations using the hydrogen bond analysis tool in VMD. The α/β tubulin interface produces the most hydrogen bonds (20 pairs) among the four tubulin complexes. Of these, 10 pairs have occupancies greater than 40% (Figure 7C). The second greatest number of hydrogen bonds is found at the β/α tubulin interface, which contains 14 hydrogen bonds. Lastly, the α/α and β/β tubulin complexes experience fewer hydrogen bonds at the interface, both in terms of the total number and the occupancy of hydrogen bonds. Together, these observations show the interaction between α and β tubulins is more stable than the interaction of identical monomers because high occupancy hydrogen bonds indicate strong binding interactions.

Understanding the high occupancy hydrogen bonds of α/β tubulin and β/α tubulin complexes is crucial for exploring the binding mechanism of tubulin molecules. The hydrogen bonds with the top three occupancies from the α/β complex include TYR210 (α-tubulin) and ASP327 (β-tubulin), THR179 (α-tubulin) and THR351 (β-tubulin), and ARG390 (α-tubulin) and GLU343 (β-tubulin). For the β/α complex, the top three hydrogen bonds are ARG391 (β-tubulin) and ASP345 (α-tubulin), GLU181 (β-tubulin) and LYS352 (α-tubulin), and GLU69 (β-tubulin) and ARG2 (α-tubulin).

To visualize the top-contributing hydrogen-bond-forming residues during the three simulations, we calculated the occupancies of hydrogen bond-forming residues and showed their occupancies in Figure 8. Residues with high occupancies that likely contribute significantly to the binding interactions are marked in dark colors. All the high occupancy residues shown in Figure 8 are in the binding interfaces.

At the binding interface of α/α tubulins, two pairs of residues form hydrogen bonds with occupancies greater than 80% from different α tubulin proteins. The first two, ARG123 and ARG64, are from one of the α tubulin surfaces. The other two, GLU297 and GLU284, are from the other α tubulin surface. On the β/β tubulin interface, residues above the 80% threshold include ASP88 from one β tubulin and ARG282 from the other β tubulin. For the α/β complex, there are 5 residues (THR73, TYR210, GLU220, THR179, and ARG390) with more than 80% occupancy from the α tubulin, and 4 residues (LYS252, ASP327, THR351, and GLU343) from the β tubulin. Notably, THR73, GLU220, LYS252, and GLU343 form hydrogen bonds with more than one residue. Lastly, for the β/α complex, two residues from α tubulin (LYS352 and ASP345), have occupancies greater than 80%. The residues that form hydrogen bonds with multiple residues and have a high occupancy will be analyzed later in combination with the results related to salt bridges, as these interactions together play a particularly critical role in maintaining the stability between the two interfaces.

### 3.6. Salt Bridges

The salt bridges with occupancies above 10% are listed in Figure 9. The salt bridge pairs between each two of the monomers in the four tubulin complexes are shown in Figure 10. The difference in the number of salt bridges formed by the α/α, β/α, and β/β tubulin complexes is small (the quantities are between three and four). Those formed by the α/β tubulin complex total seven. This finding suggests the salt bridge contributions to the interaction between α and β tubulin are likely more important for stability compared to that of the other complexes.

Combining our salt bridge analysis with hydrogen bond results, we conclude that the interaction between α tubulin and β tubulin is the strongest. It is reasonable because the growth of a single protofilament is based on tubulin, with β tubulin exposed in the plus end and α tubulin exposed in the minus end, indicating that protofilament is assembled by incorporating α/β tubulin complexes. In the α/β tubulin complex, four out of the seven salt bridge pairs formed hydrogen bonding pairs, including three with almost one-hundred percent occupancy salt bridge pairs [ASP76 (α-tubulin) and ARG46 (β-tubulin), ASP98 (α-tubulin) and LYS252 (β-tubulin), and GLU71 (α-tubulin) and LYS252 (β-tubulin)]. On the surface of α tubulin within the α/β tubulin complex, only ARG390 forms both a salt bridge (of any occupancy) and a hydrogen bond with occupancy greater than 80%. Three other residues (ASP76, ASP98, and GLU71) from α tubulin also form salt bridges and hydrogen bonds, but the hydrogen bonds for these have lower occupancies. Residues LYS96 and GLU97 play a role in salt bridge formation, but these do not form hydrogen bonds with other residues. On the binding interface of β tubulin within the α/β complex, residues LYS252, GLU343, and ARG46 are likely key in forming the complex because these also form both hydrogen bonds and salt bridges. In addition, they interact with multiple residues when forming hydrogen bonds and salt bridges.

The β/α tubulin complex has the second strongest interaction based on the comprehensive analysis of MD simulations and electrostatic feature results. On the binding interface of the complex, all salt bridge pairs form hydrogen bond pairs, including two salt bridge pairs with occupancies above 80% [GLU181 (β-tubulin) and LYS352 (α-tubulin) as well as LYS103 (β-tubulin) and GLU254 (α-tubulin)]. In particular, the residue LYS352 on the surface of α tubulin is noteworthy, as it interacts with multiple residues to form multiple salt bridges and hydrogen bonds. In addition, residue ASP345 on the surface of the α interface also has a high occupancy (94.7%) when forming hydrogen bonds, though it is not involved in the formation of salt bridges. Some residues (residues GLU254 and ARG2 from α tubulin, and residues GLU181, GLU69, LYS103, and ASP177 from β tubulin) also play an important role in the formation of salt bridges and hydrogen bonds, although the occupancy rate is not very high.

The β/β tubulin interfaces have the weakest hydrogen bond network followed by the α/α tubulin interfaces. The weaker interactions of α/α and β/β tubulin complexes are consistent with our electrostatic feature analysis, including the electrostatic potential on their binding domains, the electric field lines, and the electrostatic forces between the interfacial surfaces. However, α/α or β/β tubulin complexes have a high occupancy of salt bridges on their surfaces (both have three salt bridge pairs with close to 100% occupancies), which means salt bridges play the most significant role in holding α/α or β/β tubulin complex. For the α/α tubulin complex, these high occupancy salt bridge pairs are GLU90 and LYS280, ARG64 and GLU284, and ARG123 and GLU297. For the β/β complex, the high occupancy salt bridge pairs are ASP88 and ARG282, ASP118 and LYS297, and GLU125 and LYS336.

## 4. Conclusions

We investigated and analyzed the electrostatic interactions among tubulins for different interfaces. Four possible binding interfaces were found for tubulin/tubulin interactions. The interactions at the interfaces were analyzed utilizing various computational tools. Electrostatic features indicate that the strongest attractive forces are found in the α/β tubulin complex, while β/α complex has weaker attraction and β/β complex has the weakest attraction; The interaction between α/α tubulins is repulsive.

Furthermore, residues forming salt bridges and hydrogen bonds were identified by MD simulations. Our results show that the α/β tubulin complex has the strongest electrostatic attractive interactions followed by the β/α tubulin complex, while β/β and α/α tubulin complexes have weak attractive and repulsive electrostatic interactions, respectively. The key forces between α and β tubulins are the hydrogen bonds on the interface of the two tubulins, which is consistent with our previous work [26]. While salt bridges play the most significant role in holding α/α or β/β tubulin complex.

This study used several approaches to study the electrostatic features of microtubules based on the interactions of tubulins. It reveals the mechanisms of tubulin-tubulin interactions which demonstrates the assembly and dynamics of the microtubule. The key residues forming hydrogen bonds and salt bridges are potential drug targets for the inhibition of cancerous cells and may help researchers in the quest to develop novel therapeutic [51,52,53].

## Figures and Tables

**Figure 1 cells-12-00238-f001:**
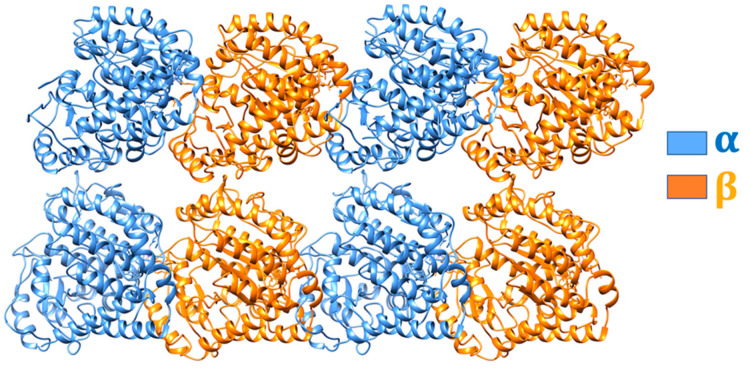
Modeled 4 × 2 array of tubulins. While α tubulins are colored in blue, β tubulins are shown in orange.

**Figure 2 cells-12-00238-f002:**
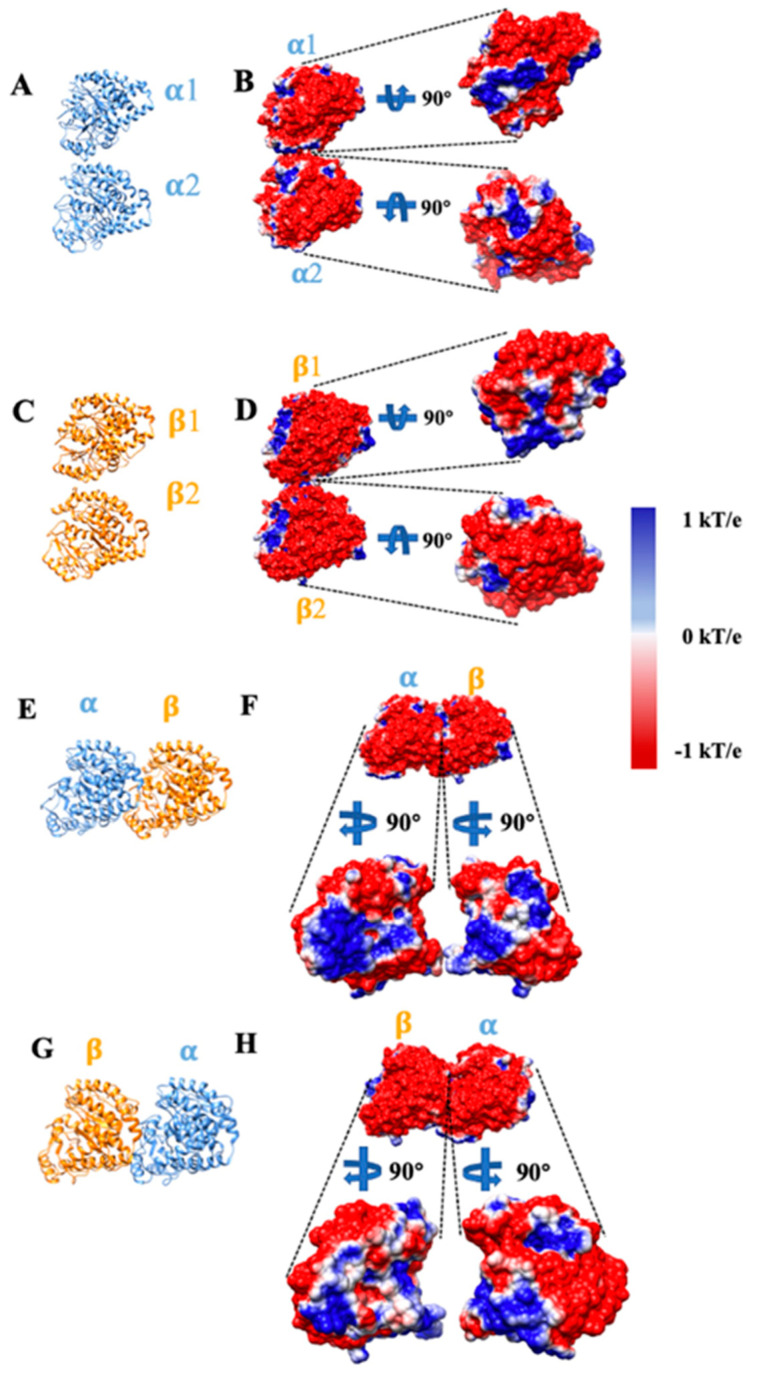
Electrostatic potential at the interfaces of tubulins (α/α tubulin complex, β/β tubulin complex, α/β tubulin complex, and β/α tubulin complex). (**A**,**C**,**E**,**G**) The ribbon structures of α/α tubulin complex, β/β tubulin complex, α/β tubulin complex, and β/α tubulin complex. All α tubulins are represented in blue, while all β tubulins are represented in orange. (**B**,**D**,**F**,**H**) The electrostatic potential surface of α/α, β/β, α/β, β/α tubulins. Negatively and positively charged regions are red and blue, respectively. In each complex, the monomers are separated by 20 Å and rotated by 90° to show the binding interfaces.

**Figure 3 cells-12-00238-f003:**
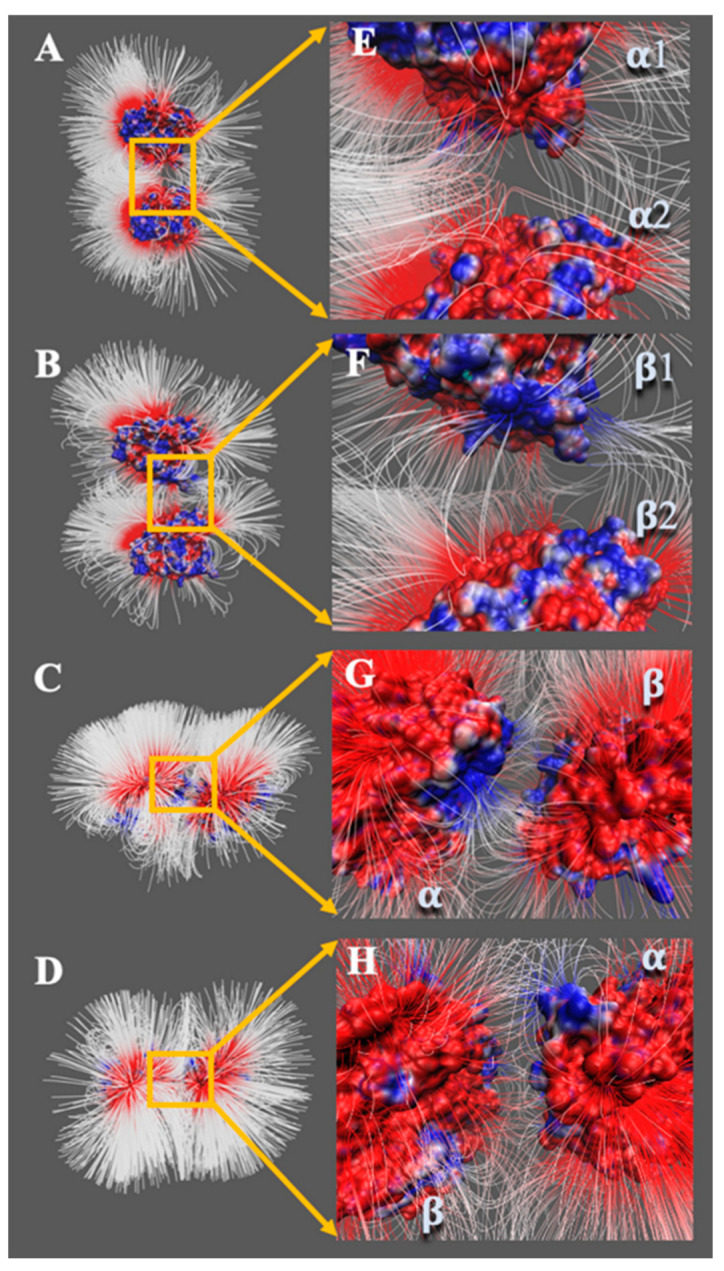
Electric field lines at the interfaces of interacting tubulins (α/α, β/β, α/β, β/α). The whole view of electric field lines of (**A**) α/α tubulin complex, (**B**) β/β tubulin complex, (**C**) α/β tubulin complex, and (**D**) β/α tubulin complex. (**E**–**H**) are the close-up views of (**A**–**D**), respectively.

**Figure 4 cells-12-00238-f004:**
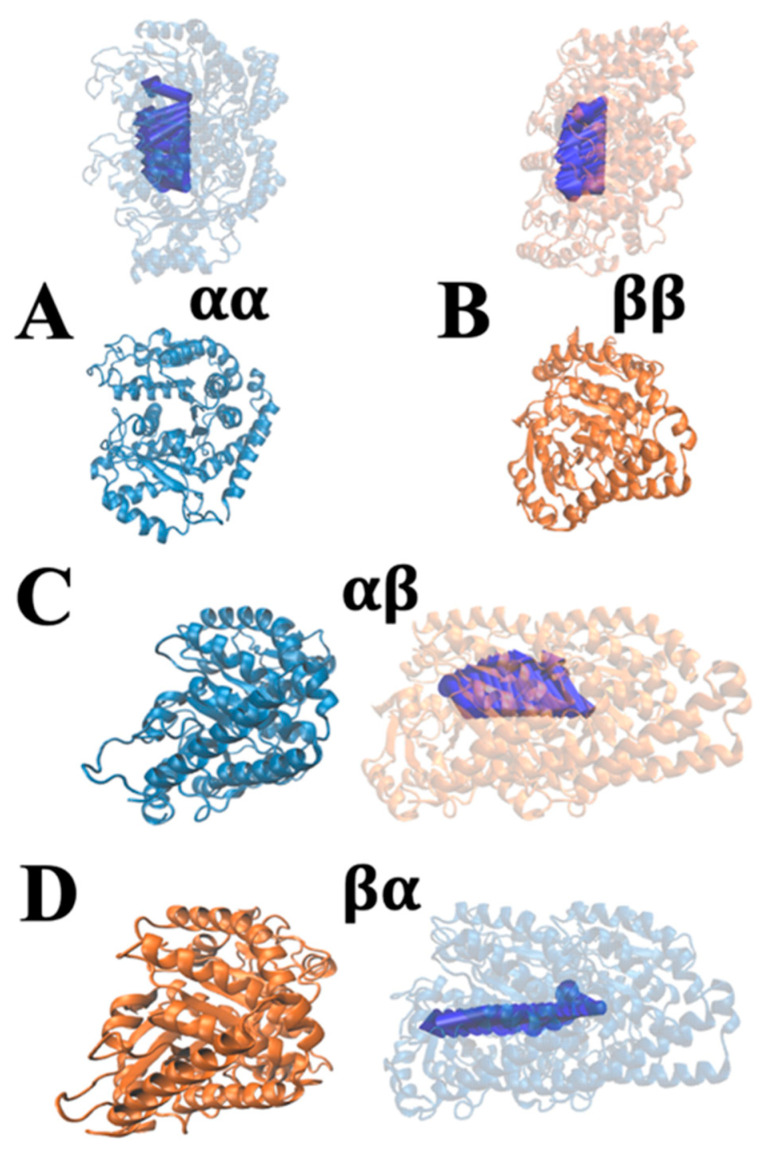
The electrostatic forces between four tubulin complexes (α/α, β/β, α/β, β/α). Tubulins are separated at distances from 14 Å to 40 Å with a step size of 2 Å, where the blue arrows show the net force directions. The electrostatic forces are shown for (**A**) α/α, (**B**) β/β, (**C**) α/β, and (**D**) β/α complexes.

**Figure 5 cells-12-00238-f005:**
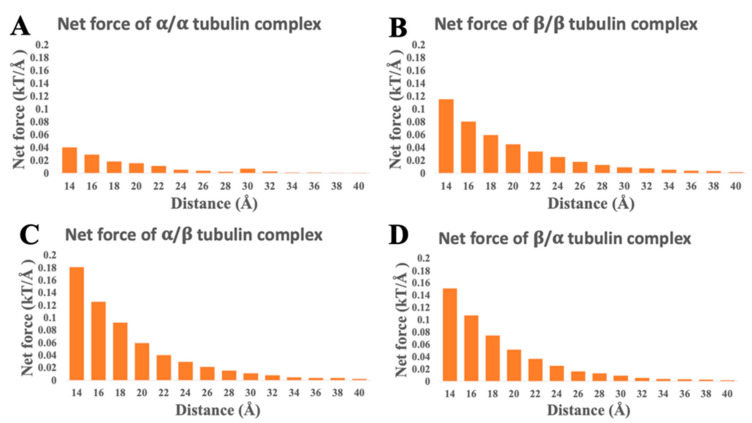
The magnitudes of electrostatic forces for tubulin complexes (α/α, β/β, α/β, β/α) at distances from 14 Å to 40 Å with a step size of 2 Å. (**A**) α/α tubulin complex. (**B**) β/β tubulin complex. (**C**) α/β tubulin complex. (**D**) β/α tubulin complex.

**Figure 6 cells-12-00238-f006:**
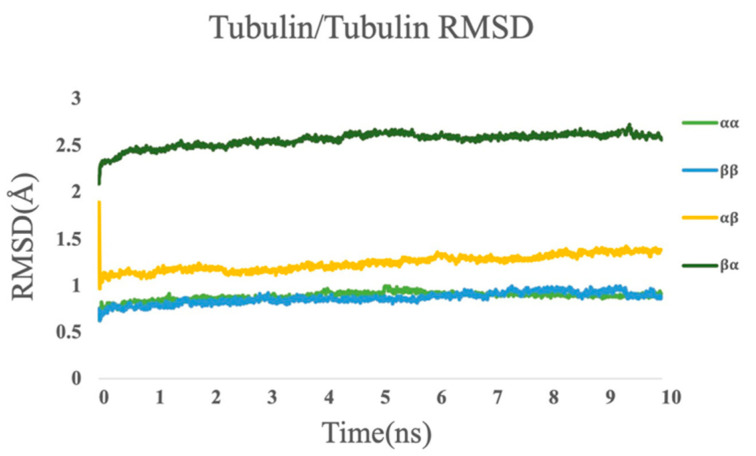
RMSD plot of tubulin complexes for α/α tubulins, β/β tubulins, α/β tubulins, and β/α tubulins.

**Figure 7 cells-12-00238-f007:**
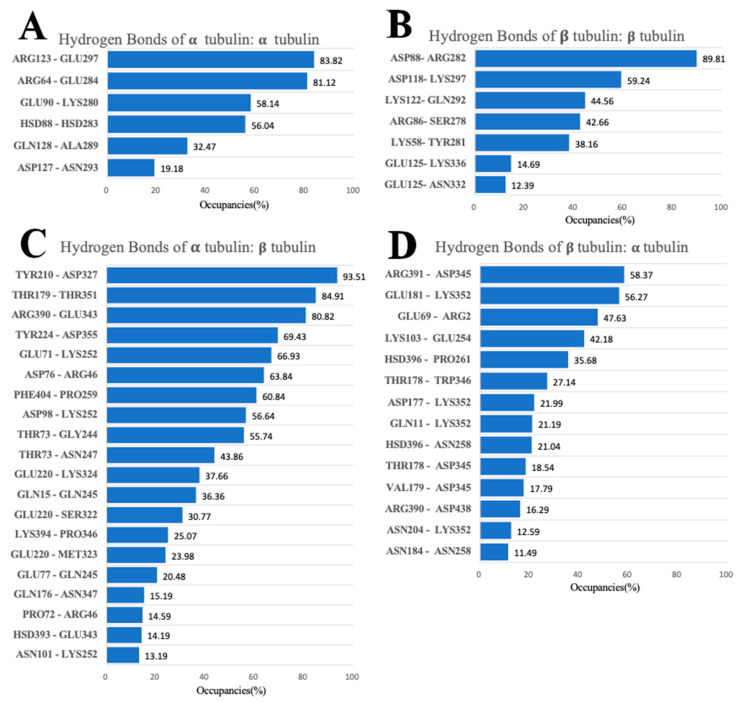
Hydrogen bonds at the interfaces of α/α, β/β, α/β, and β/α tubulin complexes with their occupancies. (**A**) α/α tubulin complex. (**B**) β/β tubulin complex. (**C**) α/β tubulin complex. (**D**) β/α tubulin complex.

**Figure 8 cells-12-00238-f008:**
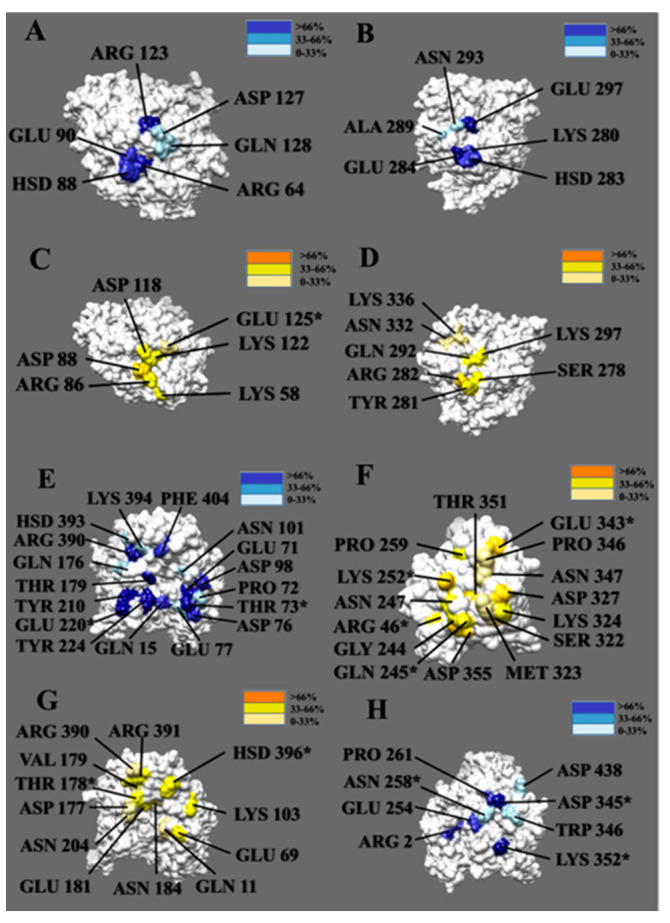
Residue distribution across the binding interfaces of tubulin subunits is color-coded based on the occupancy percentage. (**A**,**B**) Residues forming salt bridges at the interface of the α/α tubulin complex. (**C**,**D**) Residues forming salt bridges at the interface of the β/β tubulin complex. (**E**,**F**) Residues forming salt bridges at the interface of the α/β tubulin complex. The left is α tubulin while the right is β tubulin. (**G**,**H**) Residues forming salt bridges at the interface of β/α tubulin complex. The left is β tubulin while the right is α tubulin. (* indicates residues with more than one salt bridge interaction).

**Figure 9 cells-12-00238-f009:**
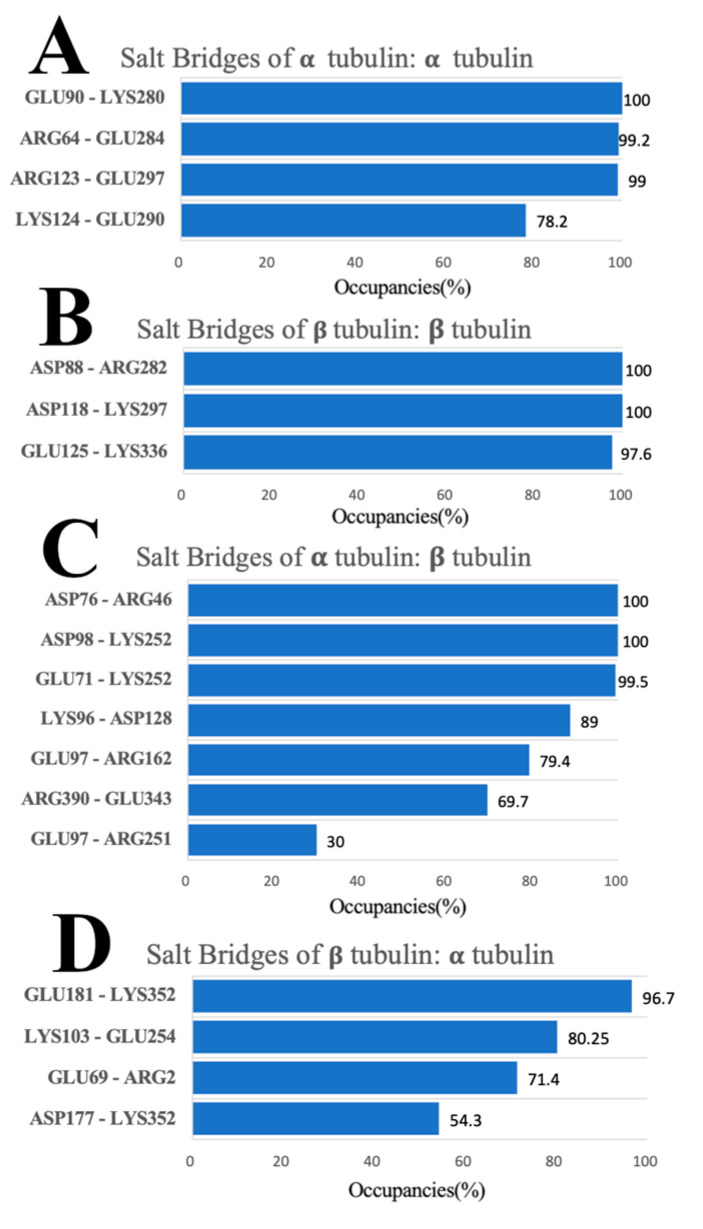
Salt bridges for α/α, β/β, α/β, and β/α tubulin-tubulin complexes with their occupancies. (**A**) α/α tubulin complex. (**B**) β/β tubulin complex. (**C**) α/β tubulin complex. (**D**) β/α tubulin complex.

**Figure 10 cells-12-00238-f010:**
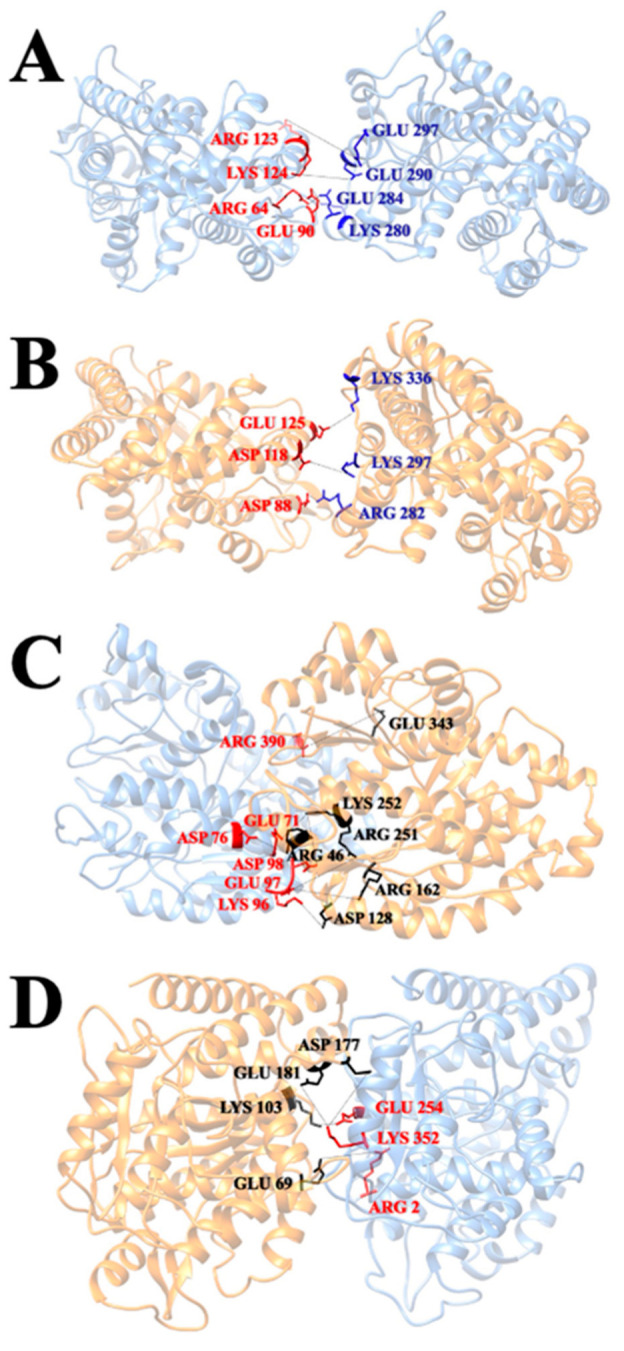
The distribution of salt bridges for α/α, β/β, α/β, and β/α tubulin-tubulin complexes with their occupancies. (**A**) α/α tubulin complex. (**B**) β/β tubulin complex. (**C**) α/β tubulin complex. (**D**) β/α tubulin complex.

## Data Availability

Not applicable.

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
