# Peer review of "A Comprehensive Study on the Electrostatic Properties of Tubulin-Tubulin Complexes in Microtubules"

_cells, 2023, doi:10.3390/cells12020238_

Round 1
Reviewer 1 Report
The manuscript entitled 'Electrostatic Interactions Play Essential Roles in Microtubule Dynamics' by Guo et al submitted to Cells employs a computational approach to assess the electrostatic interactions between tubulin monomers.
Being a wet lab cell biologist, I have limitations in assessing the scientific soundness of the methods used. Hopefully one of my fellow reviewers will be able to take care of this.
Overall, I appreciate the effort in trying to shed light on why alpha-beta dimers, but not others, are the most stable tubulin monomer pairs. However, I have a feeling that the manuscript is largely descriptive in nature. Instead of just suggesting 'hydrogen bonds and salt bridges are potential drug targets for the inhibition of cancerous cells' in the conclusion, the authors could have shown how one (or more) of the known drugs that alter microtubule dynamics, such as paclitaxel, vinblastine etc., could affect the hydrogen bonds or salt bridges of alpha-beta dimers, see https://pubmed.ncbi.nlm.nih.gov/21381049. Lastly, check for typos in the text, for eg., in the abstract I read ‘dimmers’, it should be ‘dimers’.
Author Response
Thank you for your careful review and suggestions.
- This work utilizes a series of computational approaches, such as MD simulations, DelPhi, and DelPhiForce, to study the electrostatic interactions at the binding interfaces of tubulin monomers. This work is the first detailed study on the electrostatic features of tubulins interactions. Many simulations, analyses, and calculations are included in our work. To be precise, four 10 ns simulations were performed on each of the tubulin complexes in this work to analyze the salt bridges and hydrogen bonds. To study the role of individual residues on binding interactions, we extracted one thousand frames from the last 5ns for the analysis. The efforts made include our calculation of the number of hydrogen bonds present at a distance of 3.2 Å and analysis of the salt bridges formed within 4.0 Å at each binding interface.
- Regarding the known drugs that alter microtubule dynamics: This work mainly investigates the electrostatic interactions at the binding interfaces of tubulin monomers. The known drugs that alter microtubule dynamics are interesting and rewarding. Studying drug-protein interactions needs different approaches than the protein-protein interactions. It could be the direction of our future work.
- We have corrected the typo error in the abstract and highlighted it. Thank you for your suggestions.

Reviewer 2 Report
In this manuscript, the authors used computational methods to study the various interfaces within tubulins. The emphasis was placed on electrostatic interactions. The authors calculated electrostatic force and identified salt bridges and hydrogen bonds and their occupancy. The main conclusion is that the electrostatic interactions are stronger for some interfaces compared to other interfaces.
Although the calculations and analysis are all technically correct, there is disconnection between the calculations and the topic of the paper, as reflected in the title. The paper did not address the central question, which is the role of electrostatic interactions in microtubule dynamics. Most of the conclusions drew by the authors can be easily deduced from the static structure. For example, the areas of binding interfaces between various subunits must be different. Larger binding interfaces usually correspond to stronger interactions of all kinds, including the electrostatic interactions. The analysis of only the relative interaction strength gave very little information on the flexibility of the interface, which is supposedly one of the topics of this study, as indicated in the introduction. Likewise, presenting only the occupancy of salt bridges and hydrogen bonds gave very little information on the dynamics of the interfaces and how the dynamics affect the assembly of microtubules. The lack of structural information on the dynamic binding interfaces is the major flaw of this study. If the goal was to only study the electrostatic interactions at the binding interface, the authors should change their title and part of the introduction to reflect this fact. Otherwise, the author should discuss how their study provides new insights into the binding interfaces, which have already been well defined by the static structures.
Author Response
Response: Thank you for your careful comments and suggestions, which are very useful to improve the manuscript.
We have changed the title of the manuscript to “A comprehensive study on the electrostatic properties of tubulin-tubulin complexes in microtubules”, and changed the related content about microtubule dynamics.
We agree with the reviewer that “Larger binding interfaces usually correspond to stronger interactions of all kinds, including the electrostatic interactions”. For VDW interactions, it is true larger interface results in stronger VDW attractions. But many cases show the electrostatic interactions may be repulsive even the binding interface is large. The electrostatic interaction mostly depends on the charge distribution at the interfaces. Therefore, we focused on the electrostatic interactions in this work.
We also agree that the dynamic features at the binding interfaces are crucial to study the interactions between tubulins. Therefore, four 10 ns simulations were performed on each of the tubulin complexes. In each simulation, the binding interfacial residues are free to move. The dynamics and flexibility of the interfacial residues (especially their side chains) are important to this study, because the salt bridges and hydrogen bonds are sensitive to the distance and angle of the pairwise residues. Corresponding explanation is added and highlighted in the method section.

Reviewer 3 Report
In this manuscript the authors proposed a comprehensive study on the electrostatic properties of (six different) tubulin-tubulin complexes and their role in microtubule dynamics. The authors used several computational tools and different approximations to characterize the mean electric potential, electric field, and electric force remarking the differences and similarities found between these systems. They also calculated binding sites and salt bridges. It is an important work to characterize microtubule dynamics. It is well organized and clearly written. I think that this work need to be improved to be published. I recommend the authors to resubmit the manuscript addressing the following major issues:
-The proposed study (eq. 1 in the manuscript) on mean electric potential, electric field and force only consider monovalent salt at 0.15M concentration. A realistic model for biomolecular electrostatic calculations (and microtubule dynamics studies) should include divalent ions (Mg and Ca) since they may condensate and partially neutralize the charge protein due to the high protein charge and also dominate in the diffuse layer the electrostatic screening. Thus, divalent ions cannot be omitted in these calculations.
-It is well-known that Eq.1 in the manuscript (continuum solvent model) may be inappropriate to characterize protein – protein interactions at short separation distances since electrostatic may not dominate the interaction due to the presence of short-distance interactions including ion correlations, and water crowder (explicit molecules) effects. Specifically, these other forces affect the ion distributions, therefore, the electrostatic potential of the system at short-separation distance. Thus, they cannot be omitted. Classical density functional and integral equation theories (3DRISM) might be more accurate tools for these calculations. It can also be considered modern computational tools based on molecular dynamics simulations like g_elpot: A Tool for Quantifying Biomolecular Electrostatics from Molecular Dynamics Trajectories (J. Chem. Theory Comput. 2021, 17, 5, 3157–3167). Other computational tools may be found in the article “Classical Electrostatics for Biomolecular Simulations” (Chem Rev. 2014 Jan 8; 114(1): 779–814).
-Novelty is unclear since there is a lack of comparison/discussion with previous computational work and experiments on tubulin-tubulin complexes (Introduction and discussion sections). Just to mention a few examples: “Mechanical Model of the Tubulin Dimer Based on Molecular Dynamics Simulations” J Biomech Eng. Aug 2008, 130(4): 041008 (7 pages); “Probing the origin of tubulin rigidity with molecular simulations” PNAS 05 (41) 15743-15748; “Tubulin response to intense nanosecond-scale electric field in molecular dynamics simulation” Scientific Reports volume 9, Article number: 10477 (2019); “Molecular dynamics simulations of tubulin structure and calculations of electrostatic properties of microtubules” Mathematical and Computer Modelling Volume 41, Issue 10, May 2005, Pages 1055-1070; “Tubulin Polarizability in Aqueous Suspensions” ACS Omega 2019, 4, 5, 9144–9149; “Microtubule Stability Studied by Three-Dimensional Molecular Theory of Solvation” Biophys J. 2007 Jan 15; 92(2): 394–403.
- To determine the role of electrostatics in microtubule dynamics and stability it must be included all the forces acting in the tubulin-tubulin system. The second virial coefficient calculation is key to study protein-protein interactions since it can be measured experimentally (see for example “Protein-protein interactions in concentrated electrolyte solutions” 2002 Aug 20;79(4):367-80; “Electrostatics Control Actin Filament Nucleation and Elongation Kinetics”, THE JOURNAL OF BIOLOGICAL CHEMISTRY VOL. 288, NO. 17, pp. 12102–12113, April 26, 2013). It can also bee predicted using several computational tools including DLVO-like theory for spherical particles from where the potential of mean force can be obtained. Protein-protein interactions are usually attractive a short distances and repulsive asymptotically. Something very different from the results provided by the authors in Figure 5 (when only considering electrostatic interactions).
Author Response
Thank you for the comments. The manuscript has been revised and improved according to the comments. Here is our response to the questions:
- The proposed study (eq. 1 in the manuscript) on mean electric potential, electric field and force only consider monovalent salt at 0.15M concentration. A realistic model for biomolecular electrostatic calculations (and microtubule dynamics studies) should include divalent ions (Mg and Ca) since they may condensate and partially neutralize the charge protein due to the high protein charge and also dominate in the diffuse layer the electrostatic screening. Thus, divalent ions cannot be omitted in these calculations.
Response: Thank you for the great suggestions. We completely agree with the reviewer that the divalent ions (Mg/Ca) are essential for the accurate simulation in microtubule. Fortunately, Dr. Lin Li worked as a developer of Delphi package. Therefore, we are familiar with the divalent ions function in DelPhi package to simulate the Mg/Ca effects. This function has been used in our calculations to study the tubulins. The multi-valence ion function in DelPhi has been improved several times and one reference is: Z. Jia, L. Li, A. Chakravorty, and E. Alexov, ‘Treating ion distribution with Gaussianbased smooth dielectric function in DelPhi‘, J Comput Chem. 2017 August 15; 38(22):19741979
We have modified the method section to indicate the usage of this divalent function, which has been highlighted in the revision.
Recently, our lab developed a hybrid ion treatment method which can be used to model the Ca/Mg ions explicitly, and then calculate the electrostatic features using the implicit solvent model. “Hybrid method for representing ions in implicit solvation calculations, S Sun, C Karki, Y Xie, Y Xian, W Guo, BZ Gao, L Li, Computational and structural biotechnology journal, 2021”. The detail of ion effects in tubulin-tubulin binding will be studied using the hybrid method in our next work. The details of ion effects need to be investigated separately due to the importance and complexity. We appreciate and fully agree with the points from the reviewer.
- It is well-known that Eq.1 in the manuscript (continuum solvent model) may be inappropriate to characterize protein – protein interactions at short separation distances since electrostatic may not dominate the interaction due to the presence of short-distance interactions including ion correlations, and water crowder (explicit molecules) effects. Specifically, these other forces affect the ion distributions, therefore, the electrostatic potential of the system at short-separation distance. Thus, they cannot be omitted. Classical density functional and integral equation theories (3DRISM) might be more accurate tools for these calculations. It can also be considered modern computational tools based on molecular dynamics simulations like g_elpot: A Tool for Quantifying Biomolecular Electrostatics from Molecular Dynamics Trajectories ( Chem. Theory Comput.2021, 17, 5, 3157–3167). Other computational tools may be found in the article “Classical Electrostatics for Biomolecular Simulations” (Chem Rev. 2014 Jan 8; 114(1): 779–814).
Response: Thank you for the suggestion. As a developer of DelPhi package, Dr. Lin Li has been developing and applying DelPhi to calculate the electrostatic features for many years. DelPhiForce is a specific tool of DelPhi Package developed by Dr. Li in 2016 to study the protein-protein interactions. DelPhiForce has been downloaded and used by many users and demonstrated to be very successful at protein-protein/DNA/RNA interactions. We didn’t show the details of DelPhiForce in the manuscript. Instead, two references of DelPhiForce were added. We fully understand the advantage and shortage of the DelPhi package. In this study, we used DelPhi package (including DelPhiForce) to calculate the electrostatic forces, electric field lines and electrostatic potential when the tubulins were separated 20 angstroms away from each other, to avoid clashes and computing errors when they are too closed. For the large systems such as a tubulin-tubulin dimer, we prefer to use DelPhi package to study the electrostatic interactions. Thanks for the suggestion from the reviewer, we also planned to use Density Function Theory in the future when we study more detailed interactions at a small local region, when tubulin binds with ATP. Also, thanks for letting us know the g_elpot software. We will give it a try when we work on the future research of tubulins. In this study we just used the common software, NAMD, to perform the MD simulations and salt bridge analyses.
- Novelty is unclear since there is a lack of comparison/discussion with previous computational work and experiments on tubulin-tubulin complexes (Introduction and discussion sections). Just to mention a few examples: “Mechanical Model of the Tubulin Dimer Based on Molecular Dynamics Simulations” J Biomech Eng. Aug 2008, 130(4): 041008 (7 pages); “Probing the origin of tubulin rigidity with molecular simulations” PNAS 05 (41) 15743-15748; “Tubulin response to intense nanosecond-scale electric field in molecular dynamics simulation” Scientific Reports volume 9, Article number: 10477 (2019); “Molecular dynamics simulations of tubulin structure and calculations of electrostatic properties of microtubules” Mathematical and Computer Modelling Volume 41, Issue 10, May 2005, Pages 1055-1070; “Tubulin Polarizability in Aqueous Suspensions” ACS Omega 2019, 4, 5, 9144–9149; “Microtubule Stability Studied by Three-Dimensional Molecular Theory of Solvation” Biophys J. 2007 Jan 15; 92(2): 394–403.
Response: Thank you for these wonderful suggestions. The comparison/discussion with previous computational work on tubulin-tubulin complexes have been added to the introduction section, and the corresponding revision is highlighted in the manuscript.
- To determine the role of electrostatics in microtubule dynamics and stability it must be included all the forces acting in the tubulin-tubulin system. The second virial coefficient calculation is key to study protein-protein interactions since it can be measured experimentally (see for example “Protein-protein interactions in concentrated electrolyte solutions” 2002 Aug 20;79(4):367-80; “Electrostatics Control Actin Filament Nucleation and Elongation Kinetics”, THE JOURNAL OF BIOLOGICAL CHEMISTRY VOL. 288, NO. 17, pp. 12102–12113, April 26, 2013). It can also bee predicted using several computational tools including DLVO-like theory for spherical particles from where the potential of mean force can be obtained. Protein-protein interactions are usually attractive a short distances and repulsive asymptotically. Something very different from the results provided by the authors in Figure 5 (when only considering electrostatic interactions).
Response: Thank you for your comments. We agree that the short distance forces, such as VDW force, also play significant roles in protein-protein interaction. In this study, we only focus on the electrostatic forces, especially when the tubulins are separated at a distance. For example, in figure 5, we separated the tubulins by at least 14 angstroms. We understand that when the distance gets shorter, there could be energy traps which is formed by the combination of attractive and repulsive forces. Such phenomenon is the total effect of electrostatic forces, VDW forces, polar and non-polar solvation effects, etc. But in this study, the goal is to fully understand the long-range electrostatic interactions. This is important to reveal the mechanisms of how the tubulins form a microtubule when they are far apart. For the bound structures, we performed MD simulations to study the salt-bridges to identify which residues are the important ones for holding the complex structure.

Round 2
Reviewer 2 Report
The authors have addressed my concerns. However, the authors should correct the typos before they submit the final version for publication.
Author Response
Thank you for your review and suggestions. We have corrected the typos and highlighted them in the revision.
Reviewer 3 Report
The authors addressed most of my concerns. I recommend considering the following changes for the final version of the manuscript:
-I am not questioning DelPhiForce package. While the authors fully understand the advantages and shortages of this package, I think many readers may not be aware of it. It would be worth mentioning what those are in the manuscript. It would help readers to understand what is the accuracy and what is missing in those calculations. In particular, in the current version, it is not clear that the results for the electric potential, electric field, and force are valid only for large separation distances where short interactions can be neglected.
-While in the current version, the authors mentioned previous work in the introduction, a comparison/discussion with previous results on the electrical properties of tubulins is still missed in the results and discussions section of the manuscript. This is of significant importance to highlight the novelty/relevance of the present work and, mainly, when the comparison between new predictions and experimental results is not possible.
Author Response
-I am not questioning DelPhiForce package. While the authors fully understand the advantages and shortages of this package, I think many readers may not be aware of it. It would be worth mentioning what those are in the manuscript. It would help readers to understand what is the accuracy and what is missing in those calculations. In particular, in the current version, it is not clear that the results for the electric potential, electric field, and force are valid only for large separation distances where short interactions can be neglected.
Response: Thank you for the great suggestion. We added some descriptions in the method section to help the readers understand the features and limit of DelPhi and why we separate the tubulins by 20 angstroms. This part has been highlighted in the revision.
-While in the current version, the authors mentioned previous work in the introduction, a comparison/discussion with previous results on the electrical properties of tubulins is still missed in the results and discussions section of the manuscript. This is of significant importance to highlight the novelty/relevance of the present work and, mainly, when the comparison between new predictions and experimental results is not possible.
Response: Thank you for the suggestion. We have added a comparison with previous results on the electrostatic properties of tubulins in the introduction section. This is very helpful to improve the manuscript.